# Relationships between Pulmonary Hypertension Risk, Clinical Profiles, and Outcomes in Dilated Cardiomyopathy

**DOI:** 10.3390/jcm9061660

**Published:** 2020-06-01

**Authors:** Ewa Dziewięcka, Sylwia Wiśniowska-Śmiałek, Aleksandra Karabinowska, Katarzyna Holcman, Matylda Gliniak, Mateusz Winiarczyk, Arman Karapetyan, Monika Kaciczak, Piotr Podolec, Magdalena Kostkiewicz, Marta Hlawaty, Agata Leśniak-Sobelga, Paweł Rubiś

**Affiliations:** 1Department of Cardiac and Vascular Diseases, Jagiellonian University Collegium Medicum, John Paul II Hospital, 31-202 Krakow, Poland; ewa@dziewiecka.pl (E.D.); swisniowskasmialek@gmail.com (S.W.-Ś.); akarabinowska@gmail.com (A.K.); katarzyna.holcman@gmail.com (K.H.); ppodolec@interia.pl (P.P.); magdalena.kostkiewicz@uj.edu.pl (M.K.); m.hlawaty@szpitaljp2.krakow.pl (M.H.); agata.lesniak-sobelga@cmzdrowa.pl (A.L.-S.); 2Students’ Scientific Group at Department of Cardiac and Vascular Diseases, Jagiellonian University Collegium Medicum, John Paul II Hospital, 31-202 Krakow, Poland; matylda.gliniak@gmail.com (M.G.); winiarczyk.mateusz@gmail.com (M.W.); arman47k@gmail.com (A.K.); monikakaciczak@gmail.com (M.K.)

**Keywords:** pulmonary hypertension risk, echocardiography, dilated cardiomyopathy

## Abstract

Pulmonary hypertension (PH) in patients with heart failure (HF) contributes to a poorer prognosis. However, in those with dilated cardiomyopathy (DCM), the true prevalence and role of PH is unclear. Therefore, this study aimed to analyze the profile of DCM patients at various levels of PH risk, determined via echocardiography, and its impact on outcomes. The 502 DCM in- and out-patient records were retrospectively analyzed. Information on patient status was gathered after 45.9 ± 31.3 months. Patients were divided into 3 PH-risk groups based on results from echocardiography measurements: low (L, *n* = 239, 47.6%), intermediate (I, *n* = 153, 30.5%), and high (H, *n* = 110, 21.9%). Symptom duration, atrial fibrillation, ventricular tachyarrhythmia, ejection fraction, right atrial area, and moderate or severe mitral regurgitation were found to be independently associated with PH risk. During the follow-up period, 83 (16.5%) DCM patients died: 29 (12.1%) in L, 31 (20.3%) in I, and 23 (20.9%) in H. L-patients had a significantly lower risk of all-cause death (L to H: HR 0.55 (95%CI 0.32–0.98), *p* = 0.01), while no differences in prognosis were found between I and H. In conclusion, over one in five DCM patients had a high PH risk, and low PH risk was associated with better prognoses.

## 1. Introduction

Left heart diseases (LHD) are the most common cause of pulmonary hypertension (PH) [1,2,3,4]. LHD includes both types of heart failure (HF), i.e., with reduced ejection fraction (HFrEF) and with preserved ejection fraction (HFpEF), along with primary valvular diseases and congenital heart diseases. Without doubt, among cases of LHD, the most prominent disease is HF, which alone constitutes about half of all PH cases [1,2,5,6]. It is widely accepted that PH is an important negative prognostic factor in HF; however, most of our understanding is derived from studies that are relatively outdated (e.g., lacking in current disease-modifying therapies), were mostly invasive (e.g., recruited highly selected cohorts), and small-scale (usually a few dozen HF patients) [1,2,5,6,7,8,9,10].

Despite the fact that right heart catheterization (RHC) is considered the gold standard of PH diagnosis, this cannot be carried out in a broad HF population, as HF affects 1–2% of the adult population, and more than 10% of those over 70 years of age [1,7]. Thus, merely fragmentary data on RHC in HF has so far emerged, from non-representative cohorts of selected patients with advanced HF, who are candidates for heart transplantation (HTX) or left ventricular assist devices (LVAD) [1,2,6,11]. In these studies, post-capillary PH, defined as mean pulmonary artery pressure ≥25 mmHg and pulmonary wedge pressure >15 mmHg, affects between 12% and 38% of HTX/LVAD candidates [1,12]. However, these types of findings are not truly representative of general HF populations, and surprisingly, there is little reliable and current data on PH epidemiology, pathology, and its prognostic significance in HF, in contrast to PAH [13,14]. On the other hand, neither the previous nor the current update of the European Society of Cardiology (ESC) guidelines on HF recommend routine RHC for unselected HF patients [7]. Consequently, echocardiography with a detailed assessment of right heart hemodynamics is recommended as a non-invasive and widely-available tool of investigation [1,7,15,16]. The probability of PH can be reliably assessed on the basis of the echocardiographic assessment of peak velocity of tricuspid regurgitation (TRV) and additional signs of ventricular interdependence, pulmonary artery (PA) indices, as well as inferior vena cava (IVC) and right atrium (RA) indices.

In addition, it should be acknowledged that HF is a highly heterogeneous condition; there are clear and numerous distinctions between HFrEF and HFpEF patients, and patient characteristics vary enormously even within the HFrEF group itself [7]. In Europe and the USA, the most common cause of HFrEF is coronary artery disease (CAD) and myocardial infarction (MI), which can be broadly termed ischemic-HF. Of interest here is the fact that a non-negligible proportion of HFrEF is caused by dilated cardiomyopathy (DCM). Patients with DCM are much younger (typically by 15–20 years), less burdened with comorbidities and cardiovascular risk factors, have different mechanisms of cardiac pathology, and follow a different clinical course, including the likelihood of left ventricular reverse remodeling (LVRR), etc. [17,18,19,20]. In brief, ischemic-HF and DCM-driven HF can be viewed as distinct phenomena; yet, there is little data on the epidemiology, pathology, diagnostics, and outcomes of PH in DCM.

Therefore, this study aimed to analyze the clinical, echocardiographic, and laboratory profile of DCM patients with different levels of PH risk by utilizing a recommended and state-of-the-art echocardiographic assessment. The prognostic role of PH risk in DCM was also explored.

## 2. Experimental Section

### 2.1. Study Population and Protocol

Between 2010 and 2019, we included 502 consecutive in- and out-patient DCM cases with complete baseline and follow-up data. All patients underwent detailed diagnostic work-up, including clinical evaluation, laboratory tests (morphology, creatinine, electrolytes, fasting glucose, N-terminal fragment of the prohormone B-type natriuretic peptide (NT-proBNP), and C-reactive protein), electrocardiogram (ECG), and echocardiography [7,15,16,21]. DCM was diagnosed following the current ESC recommendations, based on (1) the presence of impaired left ventricle (LV) systolic function (left ventricle ejection fraction, LVEF <45%) and LV dilation detected via echocardiogram, and (2) the exclusion of significant coronary artery disease, primary heart valve disease, congenital heart disease, and severe arterial hypertension [17,21]. To exclude coronary artery disease, patients underwent coronary catheterization or computed tomography coronary angiography. During the clinical assessment, vital signs and symptoms on admission and comorbidities were assessed. The presence of comorbidities, such as diabetes mellitus, atrial fibrillation (AF; all types included), chronic obstructive pulmonary disease, prior stroke, dyslipidemia, ventricular tachyarrhythmia (VT), and left bundle branch block, were established by medical documentation or in-hospital diagnosis. The investigations were carried out following the rules of the Declaration of Helsinki of 1975 (https://www.wma.net/what-we-do/medical-ethics/declaration-of-helsinki/), revised in 2013. All of the patients gave their consent. Prior to the study, the relevant institutional committees and the Jagiellonian University Ethical Committee approved the study (the chairperson: Prof. Piotr Thor, the protocol number: 1072.6120.171.2019, the date of approval: 27 June 2019).

### 2.2. Pulmonary Hypertension Risks

PH risk was diagnosed in accordance with the current ESC 2015 guidelines [5,6]. Briefly, patients were divided into low, intermediate, and high PH-risk groups based on the echocardiographic assessment at the peak TRV (≤2.8, 2.9–3.4, >3.4 m/s) in conjunction with the presence of at least one PH sign from at least 2 out of 3 additional categories: (1) PA indices, such as diameter or acceleration time (AcT), (2) IVC and RA indices, such as diameter and the inspiratory collapse of IVC and RA end-systolic area (RAA), and/or (3) ventricular indices (Figure 1). Importantly, there were no patients with any ventricular components; all of the patients had a smaller right ventricle (RV) than LV basal diameter, and there was no flattening of the intraventricular septum. Echocardiographic assessments were carried out at index hospital admission or during out-patient visits in stable patients or after stabilization in the case of an urgent admission.

### 2.3. Definitions of Endpoints

The primary endpoint was all-cause mortality. Information on patient status was gathered from hospital records, out-patient clinics, and via telephone contact with patients or family members between November 2019 and March 2020.

### 2.4. Statistical Analysis

All parameters are presented as mean ± standard deviation, or counts (percentages) when appropriate. All quantitative variables were tested for the normal distribution of data with the Shapiro–Wilk test. Comparisons of continuous variables between low, intermediate, and high PH-risk groups were conducted with the Kruskal–Wallis test and Dunn post-hoc test (none of the analyzed parameters demonstrated a normal distribution). The chi square test was performed for the comparison of qualitative parameters between different PH groups. The association between the parameters was investigated, and the degree of PH risk in DCM was analyzed with uni- and multi-variate logistic regression methods. All parameters differentiating patients with different degrees of PH risk were analyzed as potential PH predictors in the logistic analysis. However, TRV and moderate or severe tricuspid regurgitation (TR) were not included in our regression models due to their direct associations with PH-risk assessment.

Areas under the receiver operating curve (ROC) were calculated to assess the cut-off values of LVEF and RAA for the PH algorithm. To assess the relationship between PH risk and long-term outcome survival, a log-rank test was used, and curves were plotted using the Kaplan–Meier method. The Cox proportional hazards model was employed for the analysis of predictors for all-cause mortality. All results were considered statistically significant when their *p*-value was <0.05. The Statistica package, version 13.0 (StatSoft, TIBCO Software Inc., Palo Alto, California, USA), was used for the statistical analysis.

## 3. Results

### 3.1. Baseline Characteristics

There were 239 (47.6%) patients in the low-risk PH group, 153 (30.5%) in the intermediate-risk PH group, and 110 (21.9%) in the high-risk PH group (Table 1, Figure 2, Appendix B). The three groups were differentiated by the duration and severity of HF symptoms (NYHA class, ankle edema), urgent hospital admission, AF, prior stroke, VT, LVEF, RAA, moderate or severe pulmonary, mitral and tricuspid regurgitation (PR/MR/TR, respectively), TRV, and the presence of an implantable cardioverter–defibrillator (ICD) and cardiac resynchronization therapy (CRT) (Table 1).

### 3.2. PH Risk and DCM Patients’ Profile

Nearly all of the parameters differentiating low, intermediate, and high PH-risk groups were significantly associated with low PH risk, except for prior stroke (Table 2). The duration of symptoms, AF, VT, LVEF, RAA, and MR were found to be independently associated with low PH risk.

### 3.3. PH-Risk Impact on the Outcome

During a follow-up period of mean 45.9 ± 31.3 months, 83 (16.5%) patients died: 29 (12.1%) in the low PH-risk, 31 (20.3%) in the intermediate PH-risk, and 23 (20.9%) in the high PH-risk group (Figure 3A). Low PH-risk patients had significantly lower all-cause death rates than those in the intermediate- and high PH-risk groups (HR 0.56 (95%CI 0.33–0.94), *p* = 0.03; HR 0.55 (95%CI 0.32–0.98), *p* = 0.01; respectively). All-cause mortality was similar when comparing the intermediate and high risk groups (HR 0.99 (95%CI 0.58–1.69), *p* = 0.22). Since survival rates were similar for intermediate and high PH-risk patients (in other words, non-low PH-risk patients), we performed an additional analysis in patients stratified into just two groups: the low PH-risk group (in which 29 (12.1%) out of 239 patients died) vs. the non-low PH-risk group (in which 54 (20.5%) out of 263 patients died). Patients with non-low PH risk had significantly worse survival compared to patients with low PH risks (HR 1.79 (95%CI 1.14–2.81), *p* = 0.01) (Figure 3B).

## 4. Discussion

### 4.1. Study Findings

Although nearly half of the DCM patients were at low PH risk, over one in five displayed a high PH risk. Factors found to be independently associated with increased PH risk were a longer duration of HF symptoms, the presence of ventricular and supraventricular arrhythmias (VT and AF), more severe LV systolic dysfunction, RA remodeling, and MR. It is worth noting that DCM patients within the low PH-risk group had significantly better outcomes, while intermediate PH-risk and high PH-risk patients had significantly worse (but similar) prognoses.

### 4.2. Prevalence of Pulmonary Hypertension in LHD

#### 4.2.1. Pulmonary Hypertension Assessment in HF

As previously stated, PH diagnosis with RHC is not feasible in the general HF population and hence, not recommended. Consequently, we can obtain a more all-encompassing picture regarding PH prevalence in HF from echocardiographic studies. The traditional echocardiographic approach is to add RV systolic pressure (RVSP, estimated from the Doppler spectrum of TRV) to RA pressure (RAP, estimated from IVC diameter and respiratory collapse) [22]. However, the correlation between the echocardiographic estimation of RAP and invasive measurements has been found to be as low [23,24]. Thus, the current ESC guidelines recommends a more thorough approach when it comes to PH-risk assessment, by including not only TRV but also ventricular, PA and RA/IVC components (thereby eliminating the gross inaccuracy of RAP estimation) [1,15,16]. Despite the advent of these recently-developed enhancements, the vast majority of publications on PH in HFrEF or DCM are reliant on the now-abandoned approach traditionally used, and to the best of our knowledge, only one study has implemented the current PH-risk stratification in the general HF population [25]. This study recruited 657 patients with acute HF and reported a higher seven-month hospital readmission rate in patients with high PH risk as opposed to those at non-high PH risk. According to Carballo et al., a high probability of PH, assessed according to current ERC/ERS recommendations, is present in 18% of HFrEF patients, a finding that is similar to ours [25].

#### 4.2.2. Pulmonary Hypertension in Dilated Cardiomyopathy

To date, there have been ten studies analyzing PH in DCM (see Appendix A) [26,27,28,29,30,31,32,33,34,35]. Six of them analyzed PH by means of RHC, and four via previous echocardiographic assessment. Most of the RHC studies were small with a mean study population of 113, while the echocardiographic studies analyzed a larger DCM population, with the most prominent being from Li et al., with 1119 patients. However, the cohort analyzed by Li et al. is not representative for the general DCM population; 73% of the cohort had advanced HF with NYHA III/IV. Li et al. found that the presence of echocardiographic estimates of PA systolic pressure ≥ 40 mmHg was independently associated with all-cause mortality in DCM patients. The reported prevalence of PH in DCM varies in these studies, from 14–18% in larger studies with mostly out-patients, to 54–73% in smaller cohorts comprised of DCM patients with more advanced HF [26,27,28,29,30,31,32,33,34,35]. The highest prevalence of PH in DCM was shown by Bianco et al. in the most recent study, comprised of 87 DCM patients on optimal HF therapy. This high prevalence is probably due to the fact that the patients were not representative of the general DCM population, having been referred for HTX and LVAD. To date, there have been 5 studies analyzing the differences between PH and non-PH patients with DCM in terms of RHC, and none of these were in the context of echocardiography.

### 4.3. Pathology of Pulmonary Hypertension in Heart Failure and DCM

The development of PH during the course of HF is a staged process. Firstly, systolic (as in HFrEF) or diastolic (as in HFpEF) dysfunction causes a chronic increase in pressure in the left atrium (LA), which in turn leads to increased pulmonary venous pressure [2,3,36]. This phase can be termed a passive increase of PA pressure and is potentially reversible after correction/improvement of left heart hemodynamics (e.g., prompt initiation of HF-modifying therapies, successful revascularization in ischemic-HF or operation of mitral valve pathology, early LVRR in DCM, etc.). If, despite therapy, HF progresses, there is a decrease in LA compliance (which is frequently heralded by AF) and a further increase in PA pressure. However, at this stage pulmonary endothelial dysfunction and vasoconstriction both become apparent and eventually progress to pulmonary vascular irreversible remodeling. Thus, initial (and potentially reversible) LHD may lead to secondary and usually irreversible damage to the right heart, eventually resulting in congestion of the RA and right-side circulation overall [3].

Over the years, numerous general HF studies have reported various factors involved in this process, such as LVEF, LA diameter, mitral pathology, or right heart indices. However, little is known about PH pathology in DCM. Clearly, DCM leads to HF; however, as has been outlined above, there are crucial differences between DCM and general HF patients. Therefore, we endeavored to explore whether similar (or dissimilar) factors are significant in PH development in DCM. Based on multivariable regression analysis, we report several parameters (constituting various facets of PH pathology) that were found to be independently associated with PH in DCM. PH pathology is a long-term process, and therefore, the longer DCM (HF) lasts, the higher likelihood there is of PH occurring. As for cardiac morphology and function, we observed that LVEF, RA area, and at least moderate MR are strongly associated with PH. Out of these three, the strongest predictor is MR, which increases the probability of PH by approximately 2.7 times. Finally, AF and VT were also strong predictors of PH. As for AF, this observation (also previously reported by the authors) is not entirely unexpected, since it is strongly linked to atrial pathology. However, in the case of VT, this finding is probably related to the fact that a more diseased (remodeled and fibrotic) heart is prone to serious ventricular arrhythmias [37].

### 4.4. PH Risk and Outcome in DCM

As reported earlier, PH, as analyzed by both RHC and echocardiography, worsens the prognosis in the general HF population, as well as in HFrEF [4,8,9,10,11,25,38,39,40]. However, not much is known about its impact on outcomes in DCM. Six studies were reviewed for their findings on the impact of PH on DCM outcomes (Appendix A) [27,29,30,33,34,35]. Earlier studies have featured lower levels of HF treatments and analyses of PH in terms of RHC and previously-used echocardiographic approaches. Both these studies and ours, performed on a large DCM group on optimal HF therapy, demonstrated the significant adverse effects of PH on prognosis. According to our findings, when stratifying patients by risk into three categories (low, intermediate, and high PH risk), the Kaplan–Meier curves of intermediate and high PH risk clearly overlap. Therefore, from a practical point of view, it is more important to distinguish patients at low and non-low PH risk because this dichotomization showed very early separation in their Kaplan–Meier curves. This approach revealed significantly better prognoses in low PH-risk DCM patients in contrast to those in the non-low PH-risk group, with an approximately 45% lower death risk; there were no differences between intermediate and high PH-risk groups.

### 4.5. Study Limitations

This is a retrospective single-center study carried out on a moderately large population (the second biggest in the literature). The study population was divided on the basis of echocardiographic findings, and PH was not confirmed with RHC; nevertheless, this approach is in accordance with previous and current ESC guidelines on HF. We are the first to implement the current echocardiographic approach to the study of PH risk in DCM. We measured TRV at just one point in time, which could lead to an over- or under-estimation of the PH risk; still, patients with acute hospital admission underwent echocardiographic assessment after hemodynamic stabilization. This is only an observational study, and the attempt at pathophysiological analyses is based on prior findings, mostly in a general population with LHD.

## 5. Conclusions

Nearly half of the DCM patients under study were at low risk for PH, and one in five were at high risk. The duration of symptoms, namely atrial fibrillation, ventricular tachyarrhythmia, ejection fraction, right atrium area, and moderate or severe mitral regurgitation were independently associated with pulmonary hypertension risk. Low PH-risk patients had a significantly lower risk of all-cause death than patients with intermediate and high PH-risk levels.

## Figures and Tables

**Figure 1 jcm-09-01660-f001:**
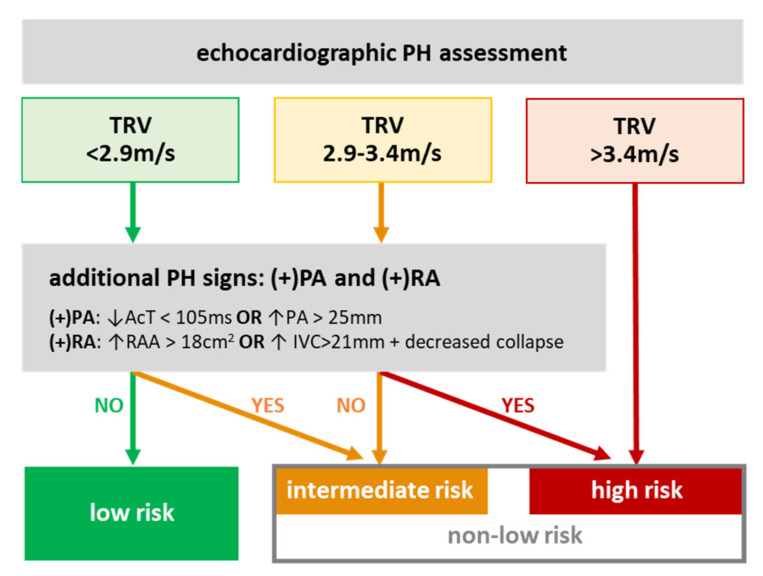
PH-risk assessment with echocardiography. Low PH-risk was defined as TRV < 2.9 m/s (or not measurable), and not more than one additional echocardiography PH sign; intermediate risk was defined as TRV < 2.9 m/s with two additional PH signs, or TRV 2.9–3.4 m/s and not more than one additional PH sign; high risk was defined as TRV 2.9–3.4 m/s with two additional PH signs, or TRV > 3.4 m/s. The additional signs for echocardiographic PH were defined as the presence of at least one sign from both categories: PA (shortened AcT or enlarged PA) and IVC/RAA (enlarged IVC with no inspiratory collapse or enlarged RA). Designations: “↓”—low, “↑”—high/enlarged, “(+)”—positive sign. Abbreviations: PH—pulmonary hypertension, TRV—peak tricuspid regurgitation velocity, PA—pulmonary artery, RA/RAA—right atrium/area, AcT—acceleration time, IVC—inferior vena cava, RV/LV—right/left ventricle.

**Figure 2 jcm-09-01660-f002:**
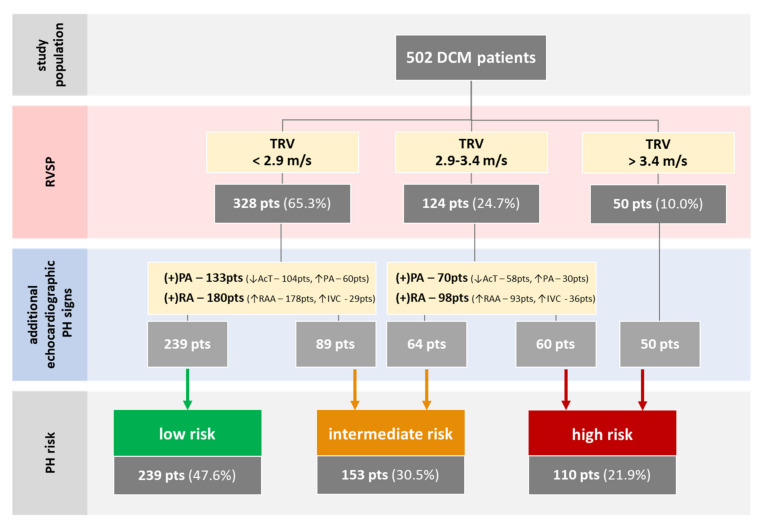
The division of the study population into three groups according to the PH-risk assessment based on Figure 1. Patients were divided according to TRV and, then in the absence of other PH signs, placed into a lower PH-risk group and in the presence of both (+)PA and (+)RA signs, to a higher PH-risk group. Designations: “↓”—low, “↑”—high/enlarged, “(+)”—a positive sign of either PA or RA. Abbreviations: DCM—dilated cardiomyopathy, pts—patients, TRV—peak tricuspid regurgitation velocity, PH—pulmonary hypertension, PA—pulmonary artery, AcT—acceleration time, RA/RAA—right atrium/area, IV—inferior vena cava.

**Figure 3 jcm-09-01660-f003:**
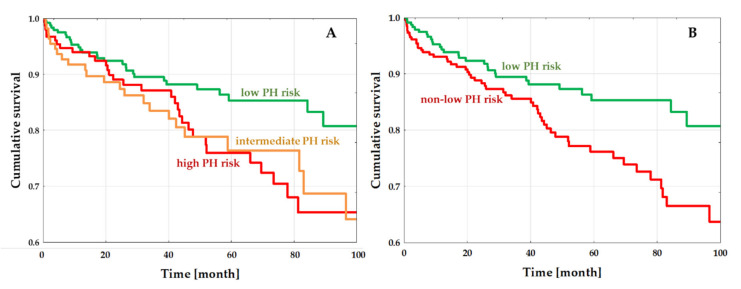
Kaplan–Meier estimates for all-cause mortality. (**A**) Study population divided into three different PH-risk groups. (**B**) Study population divided into low and non-low PH-risk groups. Low-risk patients had a significantly lower risk of all-cause death than high- and intermediate-, and non-low PH-risk DCM patients. Abbreviations: PH–pulmonary hypertension.

**Table 1 jcm-09-01660-t001:** Baseline characteristics of the study population grouped according to PH risk.

Parameters	All (*n* = 502)	Low PH Risk (*n* = 239, 47.6%)	Intermediate PH Risk (*n*= 153, 30.5%)	High PH Risk (*n* = 110, 21.9%)	*p*-Value
Age (years)	53.81 ± 13.84	52.08 ± 13.92	55.95 ± 12.86	54.62 ± 14.58	0.99
BMI (kg/m^2^)	27.69 ± 5.21	27.06 ± 5.06	28.43 ± 5.04	28.07 ± 5.60	0.10
Male (*n*, %)	403 (80.3%)	182 (76.2%)	131 (85.6%)	90 (81.8%)	0.06
Urgent admission (*n*, %)	76 (15.1%)	25 (10.5%)	27 (17.7%)	24 (21.8%)	0.0004
Symptom duration (months)	38.54 ± 56.63	30.43 ± 50.29	40.01 ± 50.53	54.12 ± 72.56	0.03 ^†^
Mean NYHA class	2.47 ± 0.90	2.2 ± 0.9	2.6 ± 0.9	2.8 ± 0.8	0.11
NYHA III/IV (*n*, %)	246 (49.0%)	83 (34.7%)	90 (58.8%)	73 (66.4%)	<0.0001
Killip 3–4 (*n*, %)	12 (2.4%)	5 (2.1%)	4 (2.6%)	3 (2.7%)	0.92
Ankle edema (*n*, %)	153 (30.5%)	52 (21.8%)	55 (36.0%)	46 (41.8%)	0.0002
DM (*n*, %)	112 (22.3%)	44 (18.4%)	40 (26.1%)	28 (25.5%)	0.13
AF (*n*, %)	160 (31.9%)	60 (25.1%)	59 (38.6%)	41 (37.3%)	0.008
COPD (*n*, %)	34 (6.8%)	13 (5.4%)	14 (9.2%)	7 (6.4%)	0.35
Prior stroke (*n*, %)	30 (6.0%)	11 (4.6%)	7 (4.6%)	12 (10.9%)	0.047
Dyslipidemia (*n*, %)	345 (68.7%)	66 (27.6%)	48 (31.3%)	43 (39.1%)	0.10
Smoker (*n*, %)					0.11
current	150 (31.1%)	81 (33.9%)	50 (32.9%)	25 (22.9%)	
previously	90 (15.9%)	33 (13.8%)	30 (19.7%)	17 (15.6%)	
SBP/DBP (mmHg)	119.62 ± 19.09/ 75.74 ± 11.98	122.46 ± 19.48/ 75.95 ± 11.96	118.32 ± 18.64/ 76.18 ± 12.53	115.22 ± 17.96/ 74.73 ± 11.33	0.09/0.26
Heart rate (bpm)	79.87 ± 19.17	77.20 ± 18.35	83.21 ± 20.34	81.04 ± 18.55	0.32
LBBB (*n*, %)	117 (23.3%)	57 (23.9%)	31 (20.3%)	29 (26.4%)	0.49
VT (*n*, %)	115 (22.9%)	38 (15.9%)	42 (27.5%)	35 (31.8%)	0.002
LVEDd (mm)	65.93 ± 10.44	63.07 ± 9.73	67.26 ± 10.28	70.28 ± 10.35	0.35
LVEF (%)	26.43 ± 10.34	29.39 ± 10.20	25.09 ± 10.24	21.84 ± 8.69	0.02 ^‡^
LVEF < 35% (%)	384 (76.5%)	161 (67.4%)	121 (79.1%)	102 (92.7%)	<0.0001
RVOT (mm)	32.66 ± 7.22	30.13 ± 5.87	34.37 ± 7.27	35.62 ± 7.97	0.27
TAPSE (mm)	18.3 ± 5.05	19.82 ± 4.96	17.50 ± 4.99	16.29 ± 4.36	0.10
LAA (cm^2^)	29.11 ± 8.79	25.06 ± 6.88	31.28 ± 8.13	34.91 ± 9.11	0.92
RAA (cm^2^)	22.97 ± 8.2	19.07 ± 6.22	25.30 ± 7.97	27.53 ± 8.46	0.008^‡^
Moderate/Severe PR (*n*, %)	23 (4.6%)	5 (2.1%)	9 (5.9%)	9 (8.5%)	0.03
Moderate/Severe MR (*n*, %)	253 (50.4%)	76 (31.8%)	97 (63.4%)	80 (72.7%)	<0.0001
Moderate/Severe TR (*n*, %)	138 (27.5%)	17 (7.1%)	52 (34.0%)	69 (62.7%)	<0.0001
TRV (m/s)	2.58 ± 1.92	2.05 ± 1.42	2.64 ± 1.64	3.40 ± 1.55	0.03^‡^
Hb (g/dl)	14.26 ± 1.58	14.38 ± 1.51	14.2 ± 1.66	14.09 ± 1.63	0.71
Creatinine (μmol/l)	94.44 ± 40.89	88.89 ± 39.99	95.76 ± 36.41	104.65 ± 46.59	0.46
LDL cholesterol (mmol/l)	2.92 ± 0.97	3.04 ± 0.96	3.01 ± 1.00	2.54 ± 0.85	0.64
Fasting glucose (mg/dl)	6.23 ± 1.92	6.00 ± 1.50	6.48 ± 2.44	6.38 ± 1.89	0.41
CRP (mg/l]	9.02 ± 23.89	5.19 ± 9.91	9.55 ± 18.87	16.60 ± 42.66	0.12
NT-proBNP (ng/mL)	3724 ± 7648	2278 ± 5207	3854 ± 5257	6655 ± 12464	0.56
ACEI/ARB/ARNI (*n*, %)	454 (90.4%)	220 (92.1%)	138 (90.2%)	96 (87.3%)	0.37
BB (*n*, %)	482 (96%)	233 (97.9%)	143 (93.5%)	106 (97.3%)	0.06
MRA (*n*, %)	437 (87.1%)	205 (86.1%)	133 (86.9%)	99 (90.8%)	0.46
Digoxin (*n*, %)	117 (23.3%)	33 (13.9%)	50 (32.7%)	34 (31.2%)	<0.0001
Loop diuretic daily dosage (mg/day)	65.22 ± 101.92	49.73 ± 91.03	71.36 ± 112.93	90.32 ± 103.23	0.57
ICD (*n*, %)	55 (11%)	10 (4.2%)	20 (13.1%)	25 (22.7%)	<0.0001
CRT (*n*, %)	16 (3.2%)	5 (2.1%)	3 (2.0%)	8 (7.3%)	0.02

Designations: ^†^*p* < 0.05 for low vs. high-risk interaction, ^‡^*p* < 0.05 for all interactions. Abbreviations: PH—pulmonary hypertension, NYHA—New York Heart Association class, DM—diabetes mellitus, AF—atrial fibrillation, COPD—chronic obstructive pulmonary disease, SBP/DBP—systolic/diastolic blood pressure, LBBB—left bundle branch block, VT—ventricular tachyarrhythmia, LVEDd—left ventricle end-diastolic diameter, LVEF—ejection fraction, RVOT—right ventricle outflow tract diameter, TAPSE—tricuspid annular plane systolic excursion, LAd—left atrial diameter (from parasternal long-axis view), LAA/RAA—left/right atrial area, PR/MR/TR—pulmonary/mitral/tricuspid regurgitation, TRV—TR peak velocity, Hb—hemoglobin, CRP—C-reactive protein, ACEI—angiotensin-converting-enzyme inhibitor, ARB—angiotensin receptor blocker, ARNI—angiotensin receptor—neprilysin inhibitor, BB—beta-blocker, MRA—mineralocorticoid receptor antagonist, ICD—implantable cardioverter-defibrillator, CRT—cardiac resynchronization therapy.

**Table 2 jcm-09-01660-t002:** Association between parameters and low PH risk.

Parameters	Univariate Regression (OR (95%CI))	*p*-Value	Multivariate Regression (OR (95%CI))	*p*-Value
Urgent admission	0.486 (0.292–0.808)	0.005	1.257 (0.585–2.698)	0.56
Symptom duration	0.995 (0.991–0.998)	0.003	0.995 (0.990–1.001)	0.03
NYHA III/IV	0.326 (0.226–0.47)	<0.0001	0.635 (0.377–1.069)	0.09
Ankle edema	0.446 (0.3–0.663)	<0.0001	0.911 (0.502–1.653)	0.76
AF	0.546 (0.372–0.803)	0.002	0.534 (0.311–0.915)	0.02
Prior stroke	0.62 (0.288–1.333)	0.22	-	-
VT	0.457 (0.29–0.72)	0.0007	0.507 (0.292–0.881)	0.02
LVEF	1.059 (1.039–1.08)	<0.0001	1.033 (1.005–1.061)	0.02
RAA	0.856 (0.827–0.887)	<0.0001	0.878 (0.842–0.915)	<0.0001
PR	0.291 (0.106–0.798]	0.02	0.591 (0.159–2.196)	0.43
MR	0.227 (0.156–0.33]	<0.0001	0.368 (0.225–0.604)	<0.0001

Abbreviations: PH—pulmonary hypertension, NYHA—New York Heart Association class, AF—atrial fibrillation, VT—ventricular tachyarrhythmia, LVEF—ejection fraction, RAA—right atrial area, PR/MR—moderate or severe pulmonary/mitral/regurgitation.

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
