# Peer review of "Relationships between Pulmonary Hypertension Risk, Clinical Profiles, and Outcomes in Dilated Cardiomyopathy"

_jcm, 2020, doi:10.3390/jcm9061660_

Round 1
Reviewer 1 Report
Ewa Dziewięcka and colleagues report their findings regarding the prognostic relevance of pulmonary hypertension (PH) risk in patients with dilated cardiomyopathy (DCM). In total, 502 DCM patients were stratified into three PH risk groups (low, intermediate, and high) based on echocardiographic evaluation. The main finding is a lower mortality risk in the low PH risk group as compared to those at high PH risk (HR 0.55). Moreover, they were able to identify some variable to be associated with PH presence in DCM (symptoms duration, atrial fibrillation, ventricular tachyarrhythmia, ejection fraction, right atrial area, and moderate or severe mitral regurgitation).
The authors present very important data on the significance of pulmonary hypertension in DCM. It should be emphasized that this is one of the largest studies of its kind. The findings are presented in a very clear and easy to understand way and the manuscript is easy to read. The discussion is carried out in an appropriate scope and important previous work is adequately included in the discussion.
Specific comments:
- Table 1 does not show (crude) statistical significances between the PH risk groups for TAPSE. However, did the authors check whether RV function has any impact on the outcome (or even modulating effects). Are additional parameters of RV performance available? It would add further value to the work if more detailed information were presented.
- Do the authors have information on myocardial inflammation (ideally from endomyocardial biopsies)? If yes, it could be of great interest to analyze whether the might be a predisposing factor for the presence of PH and to the potential modulating effect of inflammation on the outcome.
Minor points:
- Abstract, line 25: “ejection fraction” means probably “left ventricular ejection fraction”; please clarify. Please also consider using “LVEF” instead of “EF” throughout the whole manuscript in order to make clear that you refer to the left ventricle.
- From the method section it is not completely clear which variables have been considered in the regression models. It seems that all parameters listed in table 2 were used? Please clarify.
Author Response
Authors responses to Reviewers comments
Ref.: Ms. No. jcm-812022 entitled “Relationships between pulmonary hypertension risk, clinical profiles
and outcomes in dilated cardiomyopathy".
At the outset, we would like to thank the Editors and Reviewers for the great deal of time and effort expended on expertly reviewing our paper. Moreover, we are very pleased to learn that our manuscript was found to be of interest. We sincerely appreciate every comment and suggestion. All suggested changes have been made and all mistakes have been corrected in order to shape the final version of the paper (as requested all changes are highlighted with the "Track Changes" function in Microsoft Word).
We do hope you will find our revised manuscript suitable for publication in your Journal.
Answers to comments from Reviewer 1:
- Table 1 does not show (crude) statistical significances between the PH risk groups for TAPSE. However, did the authors check whether RV function has any impact on the outcome (or even modulating effects). Are additional parameters of RV performance available? It would add further value to the work if more detailed information were presented.
- As requested, we examined the possible associations between the right ventricle (RV) and outcomes in dilated cardiomyopathy (DCM). In line with the current European Association of Cardiovascular Imaging (EACVI) recommendations, we defined RV systolic dysfunction as tricuspid annular plane systolic excursion (TAPSE) < 17mm, whereas RV dilatation was defined as RV outflow track diameter (RVOT) > 33 mm. Unfortunately, other RV parameters, such as RV-S’ or fraction area change (FAC) were available only for a minority of the patients, and thus were deemed unsuitable for the purposes of meaningful analyses.
The chi2 test, Cox regression and log-rank analyses were performed, and Kaplan-Meier estimates were plotted. There were no differences in all-cause mortality between DCM patients with normal and decreased RV function (40 [14.82%] vs. 28 [19.31%], p=0.24), and no impact on the outcome results of TAPSE was found (HR 0.99 [95%CI 0.97-1.02], p=0.55) (Figure S2 - panel A). However, more patients died in the DCM group with dilated RV (51 [20.24%] vs. 32 [13.11%], p=0.03), and RVOT was found to be significantly associated with their prognosis (HR 1.04 [95%CI 1.03-1.06], p<0.0001) (Figure S2 - panel B).
Due to the fact that RV dilatation turned out to be a significant prognostic factor in univariate Cox regression, we performed an additional analysis which combined RV size and pulmonary hypertension (PH) risk. Accordingly, we stratified our patients into four sub-groups: (1) low PH risk and normal RV size, (2) low PH risk and dilated RV, (3) increased PH risk (including intermediate and high risk) and normal RV size, and (4) increased PH risk and dilated RV. Kaplan-Meier plotting revealed that patients with low PH risk and normal RV size had the lowest mortality risk (in relation to patients with dilated RV and non-low PH risk: HR 0.31 [95%CI 0.15-0.69], p=0.0007), but no differences between the other 3 groups were found (all p>0.05) (Figure S3).
- In order to present this new data, we have created a special Appendix, entitled: “RV and PH risk’s impact on outcomes in DCM”:
“Due to the importance of RV in PH, the RV modifying effect on outcomes was analysed in the context of PH risk. According to EACVI recommendations, RV function was measured using tricuspid annular plane systolic excursion (TAPSE) in apical four chamber view (A4Ch), and RV size by proximal diameter of RV outflow track [RVOT] in parasternal long axis view. RV systolic dysfunction was defined as TAPSE < 17mm, while RV dilatation was specified as RVOT > 33 mm [Rudski LG, et al. Guidelines for the Echocardiographic Assessment of the Right Heart in Adults: A Report from the American Society of Echocardiography. Endorsed by the European Association of Echocardiography, a Registered Branch of the European Society of Cardiology, And. Journal of the American Society of Echocardiography 2010, 23: 685–713. Lang RM, et al. Recommendations for Cardiac Chamber Quantification by Echocardiography in Adults: An Update from the American Society of Echocardiography and the European Association of Cardiovascular Imaging. European Heart Journal Cardiovascular Imaging 2015, 16: 233–271]. To assess the additional prognostic impact of RV size and function on long-term survival, log-rank and Cox regression analyses were performed, and curves were plotted using the Kaplan-Meier method.
There were no differences in all-cause mortality between DCM patients with normal and decreased RV function (40 [14.82%] vs. 28 [19.31%], p=0.24). However, more patients died in the DCM group with dilated RV (51 [20.24%] vs. 32 [13.11%], p=0.03). Moreover, in univariate analysis, TAPSE had no impact on outcomes (HR 0.99 [95%CI 0.97-1.02], p=0.55) (Figure S2-A), and RVOT was found to be significantly associated with prognoses (HR 1.04 [95%CI 1.03-1.06], p<0.0001) (Figure S2-B).
We also stratified patients into four groups: (1) low PH risk and normal RV, (2) low PH risk and dilated RV, (3) increased PH risk (including intermediate and high risk) and normal RV, and (4) increased PH risk and dilated RV. Kaplan-Meier plotting revealed that patients with low PH risk and normal RV had the lowest mortality risk (in relation to dilated RV and non-low PH risk: HR 0.31 [95%CI 0.15-0.69], p=0.0007), but no differences between the other 3 groups were found (all p>0.05) (Figure S3). Therefore, adding RV size to PH risk can further stratify the mortality risk for DCM patients.”
If the reviewers and editors feel that the newly-presented data is appropriate for presentation in the final manuscript version, we will be more than happy to add these paragraphs to the text.
Figure S2. Kaplan-Meier estimates for all-cause mortality. Study population divided according to RV systolic dysfunction (A) and RV dilatation (B).
Figure S3. Kaplan-Meier estimates for all-cause mortality. Study population divided according to RV dilatation and low PH risk.
- “Do the authors have information on myocardial inflammation (ideally from endomyocardial biopsies)? If yes, it could be of great interest to analyse whether the might be a predisposing factor for the presence of PH and to the potential modulating effect of inflammation on the outcome.”
- We would like to thank the Reviewer for these very pertinent comments regarding the aetiology of cardiac dysfunction in the study population. Unfortunately, only about one-fifth of the patients underwent endomyocardial biopsy (EMB). As for these EMB results, there were no cases of acute myocarditis to be found and only a small number of patients were found to have chronic or borderline myocarditis. Therefore, due to the low EMB performance in our group, we did not include EMB results in the analyses as they would be unrepresentative for the whole cohort.
In order to address the Reviewer’s query about potential associations between PH risk and DCM aetiology, we performed an additional chi2 analysis. The prevalence of various types of DCM was similarly distributed in patients with different PH risks (Table A).
Table A. Association between PH risk and DCM etiology.
Etiology [n (%)] |
All (n=502) |
Low PH risk (n=239, 47.6%) |
Intermediate PH risk (n=153, 0.5%) |
High PH risk (n=110, 21.9%) |
p-value |
Inflammatory |
48 (10%) |
19 (8%) |
17 (11%) |
12 (11%) |
0.18 |
Toxic |
68 (14%) |
34 (14%) |
24 (16%) |
16 (10%) |
|
Tachyarrhythmic |
40 (8%) |
14 (6%) |
16 (10%) |
10 (9%) |
|
Familial |
19 (4%) |
8 (3%) |
5 (3%) |
6 (5%) |
|
Other |
13 (3%) |
10 (4%) |
0 (0%) |
3 (3%) |
|
Unknown |
314 (63%) |
154 (64%) |
91 (59%) |
69 (63%) |
- “Abstract, line 25: “ejection fraction” means probably “left ventricular ejection fraction”; please clarify. Please also consider using “LVEF” instead of “EF” throughout the whole manuscript in order to make clear that you refer to the left ventricle.”
- Thank you for drawing our attention to the inexact use of the term “ejection fraction” in the text. As suggested, we have now replaced “ejection fraction” and “EF” with “left ventricle ejection fraction” and “LVEF” throughout the whole manuscript, including the abstract, the Experimental Section, Study population and protocol, the Statistical analysis subsections, the Results section, the Baseline characteristics subsection, the Discussion section, the Pathology of pulmonary hypertension in heart failure and DCM subsection, and in Tables 1, 2 and 3.
- “From the methods section, it is not completely clear which variables have been considered in the regression models. It seems that all parameters listed in table 2 were used? Please clarify.”
- The Reviewer is absolutely correct to state that the establishment of the regression model is of crucial importance to the results presented. Therefore, we included the following sentences in the Experimental Section, in the Statistical analysis subsection, and excluded them from the PH risk and DCM patients’ profile subsection and from the Results section: “All parameters differentiating patients with different degrees of PH risk were analysed as potential PH predictors in the logistic analysis. However, TRV, and moderate or severe tricuspid regurgitation (TR) were not included in our regression models due to their direct associations with PH-risk assessment.”
Answers to comments from Reviewer 2:
- Minor self-explanatory mistakes in Line 22 and line 118.
- We would like to sincerely thank the Reviewer for this valuable comment and for their very careful analysis of the text. As suggested, we corrected Line 22 in the Abstract with the following sentence: “The 502 DCM in- and outpatients records were retrospectively analysed.”, and Line 118 in the Experimental Section in the Statistical analysis subsection with “Comparisons of continuous variables between low, intermediate and high PH-risk groups were conducted with the Kruskal-Wallis test and Dunn post-hoc test (none of the analyzed parameters demonstrated a normal distribution).”
- In Table 1, it’s better to arrange columns with a year of study (Latest work in start).
- We fully agree with the Reviewer’s comment with respect to the layout and design of Table S1. Therefore, to address the Reviewer’s suggestion, we have arranged the columns in Table S1 in chronological order, with our own study analysis being placed in the last column, as follows:
Table S1. Previous studies analyzing PH in DCM.
Main author |
Romeo et al. |
Grzybowski et al. |
Rihal et al. |
Hirashiki et al. (H-2014) |
Li et al. |
Mene-Afejuku et al. |
Zhang et al.3 |
Chen et al. |
Hirashiki et al. (H-2016) |
Bianco et al. |
Own study |
Reference |
[11] |
[12] |
[13] |
[14] |
[15] |
[16] |
[17] |
[18] |
[19] |
[20] |
- |
No. of DCM patients |
104 |
144 |
102 |
256 |
1119 |
351 |
112 |
35 |
90 |
81 |
502 |
Date(s) of study |
1977-1987 |
1981-1991 |
1986-1990 |
2000-2011 |
2003-2011 |
2006-2016 |
2007-2009 |
2012-2014 |
20161 |
2016-2017 |
2010-2020 |
Study Location |
Italy |
Poland |
USA |
Japan |
China |
USA6 |
China |
China |
Japan |
Italy |
Poland |
PH identification |
R |
E |
E |
R |
E |
E |
R |
R |
R |
R |
E |
PH prevalence of PH [%], mean PASP [mmHg] or TRV [m/s] |
25mmHg |
41mmHg |
2.9m/s |
14% |
18%5 |
41mmHg |
54% |
63%, |
17% |
73%, 50mmHg |
2.6m/s |
Mean age [years] |
44 |
39 |
61 |
52 |
51 |
62 |
NA |
48 |
52 |
64 |
54 |
NYHA III/IV [%] |
49 |
73 |
34 |
103 |
73 |
NA |
81 |
80 |
1.74 |
NA |
49 |
Duration of symptoms [months] |
35 |
27 |
NA |
NA |
24 |
NA |
NA |
62 |
NA |
NA |
39 |
Mean LVEF [%] |
32 |
25 |
23 |
37 |
32 |
<40 |
32 |
29 |
30 |
26 |
26 |
Mean LVEDd [mm] |
NA |
75 |
69 |
61 |
68 |
60 |
NA |
70 |
NA |
NA |
66 |
BB / ACEi, ARB or ARNI usage [%/%] |
0/0 |
NA |
3/14 |
68/58 |
91/85 |
NA |
NA |
NA |
87/86 |
100/47 |
96/90 |
Comparison of PH and non-PH DCM patients [Yes/No] |
N |
N |
N |
Y |
N |
N |
Y |
Y |
Y |
Y |
Y |
Follow-up [years] |
3.8 |
4.1 |
3 |
4.3 |
2.8 |
0.5 |
NO |
NO |
NO |
NO |
3.8 |
Death rate [%] |
66 |
47 |
34 |
NA |
24 |
287 |
|
17 |
Designations: R – RHC, E – echocardiography, NO – no follow-up, NA – no information available. Notations: 1date of publication (conducted retrospectively), 2all patients with acute HF in last 3 months, 3available only in Chinese (analysis based on an abstract in English abstract), 4mean NYHA class, 5PASP>40mmHg, 643% of the study population was African-American, 7rehospitalization in 6 months. Abbreviations: DCM – dilated cardiomyopathy, PH – pulmonary hypertension, PASP – pulmonary artery systolic pressure, TRV – peak tricuspid regurgitation velocity, NYHA – New York Heart Association class, LVEF – ejection fraction, LVEDd – left ventricle end-diastolic, USA – the United States of America.
The updated version of the manuscript has been sent via online platform. We believe that we have responded to all the questions and appropriately corrected the paper. However, if there are any more comments and suggestions, we are more than happy to respond to all calls. We hope that the current version of the paper will be deemed of a sufficiently high standard to be accepted for publication in the highly-esteemed Journal of Clinical Medicine.
Best regards,
Ewa Dziewięcka and Paweł Rubiś
on behalf of the contributing authors
Reviewer 2 Report
In present work the author presented pulmonary hypertension in patients with heart failure that results in poorer prognosis and its role in dilated cardiomyopathy was explored. Results are well compiled and presented in the appropriate form with a detailed discussion that supports the results. I recommend it for publication with minor revision.
Comments:
Minor self-explained mistakes in Line 22 and line 118
In Table 1, it’s better to arrange columns with a year of study (Latest work in start).
Author Response

(The authors gave the same response as above.)

Round 2
Reviewer 1 Report
Many thanks to the authors for the very good revision of the manuscript and the answers to my questions. These were all adequately addressed. In my opinion it is sufficient to leave the additional table or figures in the supplement. I have no further comments.